# Numerical modeling of relative contribution of planetary waves to the atmospheric circulation

Andrey V. Koval[1,2], Olga N. Toptunova[2], Maxim A. Motsakov[2], Ksenia A. Didenko[1,2], Tatiana S. Ermakova[1,2], Nikolai M. Gavrilov[1], Eugene. V. Rozanov[3]

[1]Atmospheric Physics Department, Saint-Petersburg State University, Saint Petersburg, 199034, Russia
[2]Department of Meteorological Forecasts, Russian State Hydrometeorological University, 195196 Saint-Petersburg, Russia
[3]Physikalisch-Meteorologisches Observatorium, Davos World Radiation Centre, Davos Dorf, 7260, Switzerland

*Correspondence to*: Eugene V. Rozanov (Eugene.Rozanov@pmodwrc.ch)

**Abstract.** Using the general circulation model of the middle and upper atmosphere (MUAM), a number of numerical scenarios were implemented to study the impact of individual planetary waves (PWs) on the global atmospheric circulation, including zonal wind, temperature, and residual meridional circulation. The calculations were performed for the winter conditions of the Northern Hemisphere (January–February). The contribution to the formation of the dynamic and temperature regimes of the middle and upper atmosphere made by equatorial Kelvin waves propagating to the east, as well as atmospheric normal modes with periods from 4 to 16 days is shown. In particular, it is demonstrated that the impact of a 5-day PW and an ultrafast Kelvin wave can change the speed of circulation flows by up to 6% in the areas of their amplitude maxima. At the same time, this effect can be significantly enhanced in certain periods of time. The presented research results are important for a deeper understanding of the mechanisms of large-scale atmospheric interactions. Despite the obviousness and simplicity of the problem, such work has not been carried out at the moment.

**Keywords**: planetary waves, normal atmospheric modes, residual meridional circulation, numerical simulation, atmospheric dynamics

## 1 Introduction

Planetary waves (PWs, known as Rossby waves) are large-scale variations in the hydrodynamic parameters of the atmosphere (wind, temperature, density), which are formed due to the potential vorticity conservation. The horizontal distribution of PWs is determined by the counteraction of the meridional gradient of the Coriolis force and the meridional displacements of the jet streams. According to the classic theory (e.g., Holton, 1975), a number of waves fit along the latitude circle, determining the zonal wave number. The amplitudes of PWs increase due to a decrease in the density of the atmosphere, when they propagate from their sources in the troposphere. In the middle and upper atmosphere these disturbances become an important driver of the atmospheric circulation. One of the important features of planetary waves is their active interaction

with the mean flow causing transfer of energy and momentum. This feature was reflected in the formulation of the generalized Eliassen Palm theorem (Eliassen and Palm, 1961). PWs can provide a significant acceleration of the background flow in the middle atmosphere when dissipating. This acceleration is comparable to the acceleration associated with gravity waves and atmospheric tides (e.g., Pogoreltsev, 1999).

Another important feature of PWs, which explains the need for their comprehensive study, is that they are a link between different atmospheric layers and regions. The PWs can contribute to the signal propagation from the quasi-biennial oscillation (QBO) of the equatorial zonal wind into the thermosphere (Koval et al., 2022a,b) and from the equatorial region to the extratropical region (Holton & Tan, 1980). The ability of PWs to be reflected downward at the heights of the lower thermosphere, due to changes in vertical temperature gradients associated with solar activity cycle, can also have a significant effect on the dynamic and temperature regimes of the middle atmosphere (Koval et al., 2018a).

According to the so-called "downward control principle" (Haynes et al., 1991), PWs are the main driving force of meridional extratropical circulation (see also Holton et al., 1995). Due to its global nature, meridional circulation is considered to be the most important mechanism of dynamic interaction between different layers and regions of the atmosphere, affecting the transport of aerosol, atmospheric gases and, consequently, the composition of the atmosphere. Changes in the meridional circulation can affect the ozone layer behavior. The state of the ozone layer has attracted increased attention due to global ozone depletion (e.g., Newman et al, 2009). PWs are the main factor in the development of sudden stratospheric warming (Schoeberl, 1978; Nath et al., 2016).

A lot of studies are currently dedicated to the PWs having different periods and zonal wavenumbers. For example, numerical simulations of PWs influence were discussed in Liu et al. (2004); Chang et al. (2014); Wang et al. (2017); Forbes et al. (2018; 2020); He et al. (2020) and many others. Ground based radar measurements were presented by Clark et al. (2002); Jiang et al. (2008); Pancheva et al., (2008) and satellite measurements by Day et al. (2011); Forbes & Zhang (2017); Pancheva et al., (2018); Merzlyakov et al. (2013), as well as processing of reanalysis data/weather forecasting system by Sassi et al. (2011); Qin et al. (2021), etc.

In this paper we considered the relative contribution of various PW modes to the formation of the global atmospheric circulation using the unique opportunity that numerical modeling gives us. In order to further understand the nature of large-scale atmospheric dynamics, we carried out a number of numerical experiments to quantify the sensitivity of the zonal wind and temperature fields, as well as meridional circulation components to the switching on/off sources of various PW modes in the model. Despite the obviousness and simplicity of the problem, such work has not been carried out at the moment. Unfortunately, there is no universal way to study the impact of all Rossby waves, each wave has its own characteristics, depending, in particular, on the season, the impact of large-scale processes such as quasi-biennial oscillation of the equatorial zonal wind, El-Nino southern oscillation, etc. Therefore, we have chosen only a part of the PW spectrum, the amplitudes of which are maximized during the boreal winter.

## 2 Methodology

**The MUAM model**. Planetary waves are studied using the Middle and Upper Atmosphere Model (MUAM, Pogoreltsev et al., 2007). MUAM is a three-dimensional nonlinear mechanistic model of the general atmospheric circulation at heights from the surface to the F2 ionospheric layer (up to 300-400 km). This is one of the most promising and modern models of atmospheric wave dynamics, which makes it possible study the processes in the middle and upper atmosphere, as well as their interaction with lower levels (see, for example, Gavrilov et al., 2018; Ermakova et al., 2019; Koval et al., 2018a, b; 2022a,b; Medvedeva et al., 2019). One of the advantages of MUAM is that it allows us not only to analyze the amplitudes of planetary waves, but also to associate them with various generating sources. The log-isobaric height $x = -H \times ln(p/p_s)$ is used as the vertical coordinate in MUAM, where p is the pressure in hPa, $p_s$ is the surface pressure, and H is the pressure scale height. The latitude and longitude spacing of horizontal grid of the model is 5.625º x 5º. A version of the model with 56 vertical levels is used, covering a vertical range from the Earth surface to about 300 km. The time integration step is 225 s.

The MUAM radiation module takes into account atmospheric net radiative heating due to solar and infrared irradiance. The thermosphere includes parameterization of heating in the extreme ultraviolet band. Ion drag, molecular and turbulent viscosity and thermal conductivity are included as well. The model provides the possibility of planetary waves' excitation near the Earth's surface. The possibility of changing the albedo of the underlying surface is available. Weather changes and cloudiness in the troposphere are not simulated. The MUAM uses three parametrizations of gravity waves with different phase velocities, including orographic waves. For further description of the processes involved in the current version of the model, please refer to Koval et al. (2022a).

The main parameters simulated by the MUAM include 4-dimensional fields of the zonal, meridional and vertical velocity components, geopotential height, and temperature with time step of 2 h. By the MUAM initialization, zonal mean climatological distributions of the geopotential height and temperature are set with the lower boundary conditions at the 1000 hPa isobaric level. These distributions were obtained using the reanalysis MERRA-2 data (Gelaro, et al., 2017) and averaged over 20 years (from 2000 to 2019) for January-February.

Since the MUAM does not reproduce tropospheric weather, the sources of the westward propagating PWs (atmospheric normal modes, NMs) and the eastward PWs (Kelvin waves) in the MUAM are specified using additional terms in the heat balance equation, having the form of time-dependent sinusoidal harmonics with zonal wavenumbers *m = 1..3*, and periods matching to simulated PWs. To specify the latitudinal structure of the PW components, the corresponding Hough functions obtained using the method described by Swarztrauber and Kasahara (1985) are used. PW periods are equal to the resonant response of the atmosphere to the wave action at the lower boundary (Pogoreltsev, 1999). Westward propagating NMs (1.1), (1.2), (1.3), and (2.1), (2.2) in the classification proposed by Longuet-Higgins (1968) are considered. They have periods of about 5, 10, 16 days with a zonal number of 1, and about 4 and 7 days with a zonal number of 2. In addition, eastward propagating ultrafast Kelvin wave (UFKW, having period of about 3.5 days, a zonal number of 1) are studied. In addition to the mentioned PWs, MUAM also includes sources of slow and fast Kelvin waves (*m=1*), and quasi-two-day wave

(*m=3*). However, their amplitudes and contribution to the global circulation during the boreal winter are weak, so they are beyond the scope of this study.

The spatial resolution of the model is relatively coarse, however, as the previous studies have shown, this resolution is more than enough to resolve global atmospheric oscillations, including tides (e.g., Suvorova & Pogoreltsev, 2011; Shevchuk et al. 2018; Didenko et al., 2021) and planetary waves (e.g., Gavrilov et al., 2018; Koval et al., 2018a,b; 2022a,b and references therein). Very important drivers of the atmospheric circulation are gravity waves (GWs). Naturally, the GWs (of orographic and non-orographic origin) cannot be resolved by the MUAM, so parameterizations are used to involve their dynamic and thermal effects. There are three of them in model. For GWs having small phase speeds (5-30 m/s) a parameterization by Lindzen (Lindzen, 1981) is implemented. For faster waves with phase speeds of 30-125 m/s, which are in particular important in the thermosphere, a version of the spectral parameterization proposed by Yigit and Medvedev (2009) is applied. The parameterization uses 15 GW spectral components uniformly distributed within the period range from 40 min to 3 h. A third parameterization implemented into the MUAM is responsible for accounting of stationary GWs of orographic origin (Gavrilov and Koval, 2013).

**Residual meridional circulation**. A significant problem when considering meridional flows in the framework of the classical Eulerian approach (i.e., with zonal averaging of meridional and vertical circulation flows) is that, in the equations of dynamics, the wave sources of momentum and heat are compensated by advective flows of momentum and heat (Charney and Drazin, 1961). This feature does not allow one to isolate and analyze the wave action on the mean flow. At the same time, in the continuity equation for long-lived gas components, there is a compensation of wave and mean flows. Thus, the use of the Eulerian mean meridional circulation is inefficient for calculating mass transfer and long-lived gas species and analysing wave-mean flow interaction. A thorough analysis of this topic was made by Butchart (2014). In this study, the Transformed Eulerian Mean (TEM) approach, introduced by Andrews and McIntyre (1976), was used to diagnose the impact of PW on the mean flow. The TEM approach is based on consideration of the components of the mean residual meridional circulation (RMC), which is a superposition of eddy and advective mean transport. Formulas for calculating the RMC components are presented, for example, by Koval et al. (2022a). The time-averaged RMC represents the net average movement of air masses and, therefore, in contrast to the conventional mean Eulerian circulation, it approximates of the average advective movement of atmospheric species.

**Scenarios of model experiments**. A series of numerical experiments (model runs) was carried out for January-February to identify the influence of various wave components on the variability of the global circulation and the RMC. The scenarios of the model runs are presented in Table 1: a reference run of the model (#1) was carried out to calculate the atmospheric circulation with the inclusion of all sources of the considered PWs, and other runs were performed with the sources of individual waves turned off. Designations of 4DW, 5DW… mean PWs having periods of 4, 5 days and others. UFKW means ultrafast Kelvin wave. The PW amplitudes were obtained using the longitude-time Fourier expansion into the first 4 harmonics applied to the geopotential height fields. Next, an approximation was carried out using the least squares method to the given oscillation periods.

**Table 1. Scenarios of model calculations, including different PWs.**

| runs | 4DW | 5DW | 7DW | 10DW | 16DW | UFKW |
|------|-----|-----|-----|------|------|------|
| 1 | + | + | + | + | + | + |
| 2 | + | + | + | + | + | |
| 3 | + | | + | + | + | + |
| 4 | + | + | + | | + | + |
| 5 | + | + | + | + | | + |
| 6 | | + | + | + | + | + |
| 7 | + | + | | + | + | + |

**3 Amplitudes of planetary waves**

Fig. 1 shows the amplitudes of geopotential height variations due to the observing planetary waves for January-
February. The wave amplitude according to the results of the initial model simulation with the inclusion of sources of all
considered PWs (run #1) is presented on the left side. For comparison, the right panels show the amplitudes of these waves for
the model simulations with each wave source turned off (see scenarios in Table 1).

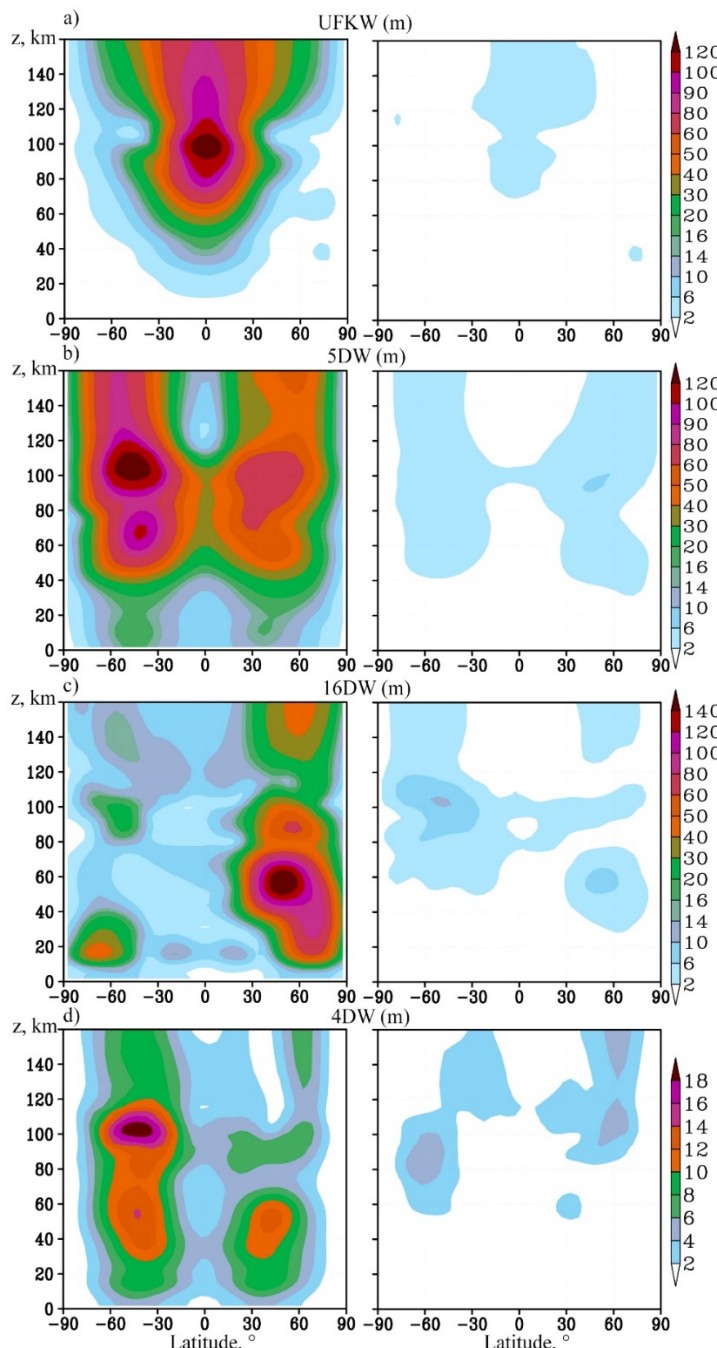

Figure 1. Amplitudes of variations of geopotential height (m) with the source of the respective PW in the MUAM been turned on (left panels) and off (right panels) for the following PW modes: a) Ultrafast Kelvin wave, b) 5-day PW, c) 16-day (all with a zonal wave number m =1); d) 4-day (with m=2). Note that the color scale is different for different panels.

The amplitude of eastward propagating UFKW (a period of about 3.5 days) is shown in Fig. 1a. Kelvin waves are

localized in the low latitude region unlike classic atmospheric NMs, the horizontal structure of which is caused primarily by

the action of the Coriolis force weakening them near the equator. The UFKW is mainly excited by the tropospheric source

specified in the MUAM. Its generation by internal atmospheric interactions is relatively weak (compare the left and right

panels of Fig.1a). The westward propagating NMs, shown in Fig. 1b-d, have maxima in the middle latitudes of both

hemispheres. Waves with larger phase velocities (4-d and 5-d NMs) can propagate in both hemispheres (Fig. 1b and 1d), while

slower waves predominantly propagate through the eastward wind structures of the winter (in our case – the Northern)

hemisphere (Fig.11c). This is due to propagation barriers of these waves occurring when their phase velocity is less than the

westward zonal jet stream in the summer stratosphere and mesosphere (see, for example, Charney and Drazin, 1961). The

presence of these barriers is also confirmed by the calculation of the refractive index of the atmosphere for the PWs considered.

According to Matsuno (1970), PWs propagate along waveguides: regions of positive refractive index. Our calculations showed

that in the Southern Hemisphere the waveguide for 10- and 16-day waves is interrupted, preventing their direct upward

propagation. These waves propagate to the Southern Hemisphere from the Northern one, crossing the equator in the

stratosphere, as was shown, for example, in the study by Koval et al. (2018a).

Fig. 1 shows the deficiency of waves generation in the middle atmosphere inside the model, and the PW amplitudes

with the sources turned off (right panels) do not exceed a numerical noise level. An exception is the maximum amplitude of

16-day PW in the right Fig. 1c, which is formed at latitude near 60° S and altitude of about 100 km. When the tropospheric

source is turned off, this maximum of geopotential height reaches 15 m in the right panel of Fig. 1c, whereas it is about 24m

for the turned-on wave source (the left panel of Fig.1c). This reveals an interesting effect of 16-day PW generating by internal

atmospheric sources was discovered. The main source of the 16-day wave generating in the southern lower thermosphere in

the MUAM may be elucidated by the nonlinear interaction of the 5- and 4-day waves, whose amplitudes have maxima in the

same latitude-altitude region in the left panels of Fig. 1b and 1d. Therefore, further study of this phenomenon is required.

A detailed comparison of the MUAM-simulated PW amplitudes for January-February with satellite and radar

observations, also with reanalysis data was carried out. For example, the amplitudes of PWs in the geopotential field calculated

according to NCEP/NCAR reanalysis data at 10 and 30 hPa pressure levels were presented in the study by Pancheva et al.,

(2008). The values of these amplitudes agree with our results. The calculated PW amplitudes in geopotential height according

to the MERRA-2 reanalysis data and averaged over the years used for the initialization of MUAM have also similar value and

structure to the simulated one's. Additionally, Yamazaki et al. (2021) presents the distributions of 4-day PW amplitudes

according to measurements of geopotential height using Microwave Limb Sounder on Aura satellite, the structure of which

corresponds to our calculations. Whereas, the presented values of the PW amplitudes may differ significantly, which is

primarily due to the fact that the data for individual specific days are presented in the specified article. The data from the global

numerical weather forecasting system (NOGAPS-ALPHA) is used by Sassi et al., (2012) to calculate structures of geopotential

height variations by atmospheric NMs. These structures are similar to our distributions. In addition, the 5-day wave

amplification in the southern mesosphere similar to the one demonstrated in the left Fig. 1a is shown. For a more detailed

analysis of the simulated PWs, in order to compare with the published data, the amplitudes of temperature variations by PWs were also calculated. The simulated 5-day PW and UFKW in temperature field were compared, in particular, with the wave amplitudes calculated from TIMED/SABER temperature data (Pancheva et al., 2010). The amplitude values accordance (up to 6 K at the MLT height for January for 5-day PW at the mid-latitudes of both hemispheres, for UFKW - at the equator) and the spatial distribution accordance of PW across latitudes were found. Moreover, the simulated PW amplitudes correlate in magnitude and spatial distribution with the respective waves obtained in a number of studies (Pancheva et al., 2008, 2009; Forbes et al., 2017; Pedatella & Forbes, 2009; Huang et al., 2017).

**4 Relative PW contribution to the general atmospheric circulation**

The residual meridional circulation (RMC) was calculated to analyze the changes in atmospheric circulation caused by various PWs for each MUAM simulation scenario presented in Table 1, with all PW sources turned on for comparisons with model runs at turned-off sources of particular wave modes. The RMC structure should be sensitive to the PW impact as it is a combination of advective and wave-induced eddy components. The latter is driven primarily to PWs according to the "Downward control principle" (Haynes et al., 1991). Fig. 2 shows the RMC components and temperature averaged over January-February for model calculation No. 1 (all PW sources included) and differences in these fields due to turning off each of analysed PW mode. Respective zonal-mean zonal wind increments are shown in Fig. 3. Simulated zonal-mean wind (Fig. 3a) and temperature (Fig. 2a) correlate with those obtained with the empirical models HWM-14 (Drob et al., 2015) and NRLMSIS 2.0 (Emmert et al., 2020), also with a semiempirical wind model by Jacobi et al. (2009) and with the MERRA-2 reanalysis data.

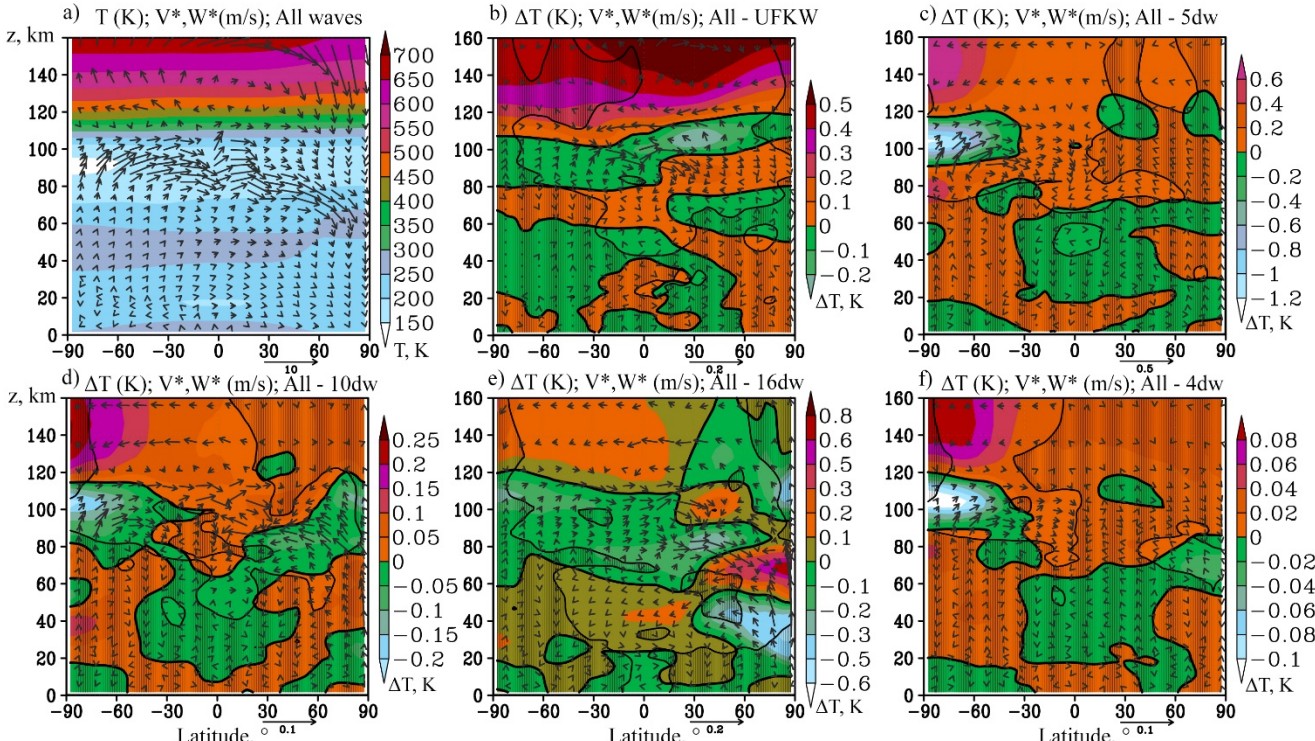

Figure 2. a) RMC components (arrows, m/s, vertical component multiplied by 200) and mean zonal temperature components (colours, K) for January-February with all PW sources turned on; b-f) increments in RMC and temperature due to switching off sources of PW: UFKW, 5-, 10-, 16- and 4-day waves, respectively. Shaded areas show insignificant temperature and/or RMC increments at 95%.

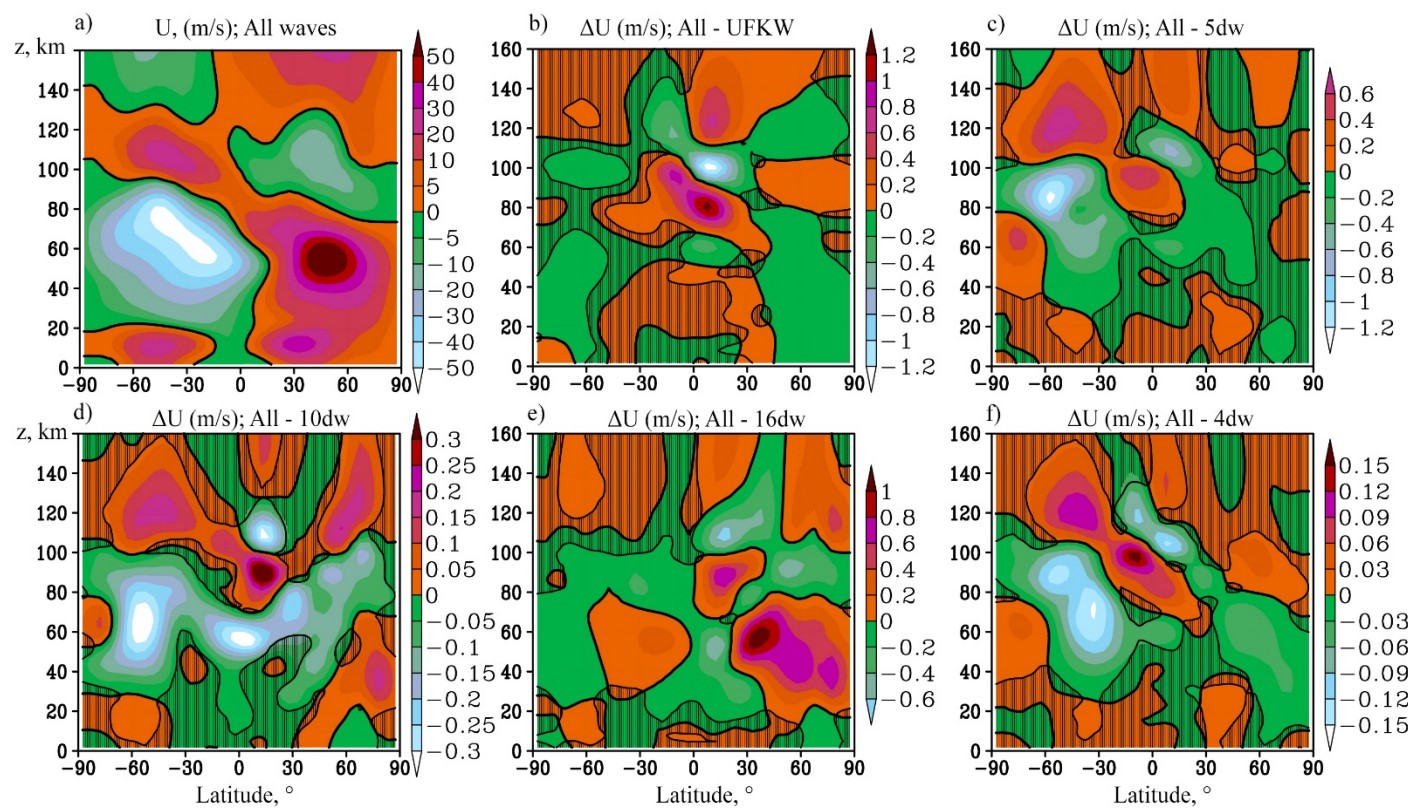

**Figure 3. a) zonal wind components (colours, m/s) for January-February with all PW sources turned on; b-f) increments in zonal**
**wind due to switching off sources of PW: UFKW, 5-, 10-, 16- and 4-day waves. Shaded areas show insignificant wind increments at**
**95%.**

Fig. 2 and 3 show influence of turning off each individual PW to the zonal-mean temperature and zonal wind. The
main impacts are usually localized in the regions of maximum PW amplitudes. The greatest contribution to the circulation
change is made by 5-day PW. The main differences in Fig. 2c occur in the southern lower thermosphere, which correspond to
a RMC strengthening in a layer between 80 and 120 km after switching on 5-d PW tropospheric source. The acceleration of
zonal wind (eastward above 100 km, and westward below) is observed in the same region in Fig. 3c. This effect is primarily
explained by the convergence of the Elissen-Palm flux (EP) in this region. The acceleration of the RMC there leads to the
lifting up of a warmer air and warming of the atmospheric layer between 60 and 90 km, as well as to the acceleration of air
transport from the coldest region of the atmosphere (about 90 km, at latitudes from the South Pole to 60° N), which leads to
the cooling of the atmosphere above this layer. In addition, in the circumpolar southern stratosphere, at a level of about 60 km,
there is deceleration of the zonal wind, which, on the contrary, is associated with the EP flux divergence. The described changes
in RMC and zonal wind between 60 and 120 km can reach values up to 6% forming a significant contribution to the atmospheric
circulation from only one wave. Relative changes in RMC components and zonal wind are presented in Figs.S1b-S3b in the
supplemental information.

The maximum UFKW amplitude is located at 100 km in the equatorial region (see Fig. 1b). Then the wave propagates

higher, gradually attenuating. Its contribution to the circulation flows changes is also maximized in this region and exerted

mainly in the strengthening of the zonal wind (Fig. 3b) and the RMC (Fig. 2b). Similar to 5-day PW, the RMC increments can

reach up to 5-6% as it is shown in Figs. Figs.S1a and S2a. Fig S3a shows that zonal mean wind changes in the equatorial

region, between 80 and 120 km can exceed 10% in areas where wind values are greater than 5 m/s. The UFKW impact in the

100-120 km layer leads to cooling in the Northern Hemisphere caused by a slowdown in meridional transport and additional

updrafts causing adiabatic cooling.

The impact of the 16-day wave on the circulation, as shown above (Figs. 2e and 3e), is comparable in value with 5-

222    day PW and UFKW, however in has different structure. Maximum PW amplitude occurs in the stratosphere of the Northern

Hemisphere, and its contribution to atmospheric circulation is observed in this region. Figure 2e shows that introduction of 16-

224    day wave leads to cooling of the layer below 50 km and heating of the overlying layer. The temperature changes here are

explained by the change in the RMC components: in particular, the acceleration and weakening of the RMC descending branch

contributes to adiabatic heating and cooling, respectively. This is accompanied by acceleration of the zonal wind (Fig. 3e),

directed in this region to the east (Fig. 3a). Statistically significant changes in circulation components may reach 6% in the

high-latitude stratosphere as shown in Figs. S1d and S2d. Below, in Figs. 5 and 6 it is shown that action of the 16-day PW may

be stronger than 5-day PW and UFKW at certain points in time.

10- and 4-day PW make a smaller contribution to the dynamic and thermal regime of the atmosphere. Specifically,

the structure of the 10-day wave in the middle atmosphere is similar to the structure of the 16-day one: the amplitude maximum

is observed in the northern stratosphere, but due to the higher phase velocity, its waveguide in the southern middle atmosphere

is wider. Propagating in the Southern Hemisphere, it contributes to the zonal wind acceleration up to heights of 140 km (Fig.

3d) and to the respective temperature changes. A faster 7-day wave, like 5-day wave, is able to propagate along waveguides

in both hemispheres. Generally, the 10- and 7-day PW contributions cause the same effects as the 5-day one described above,

although they are much weaker in this region.

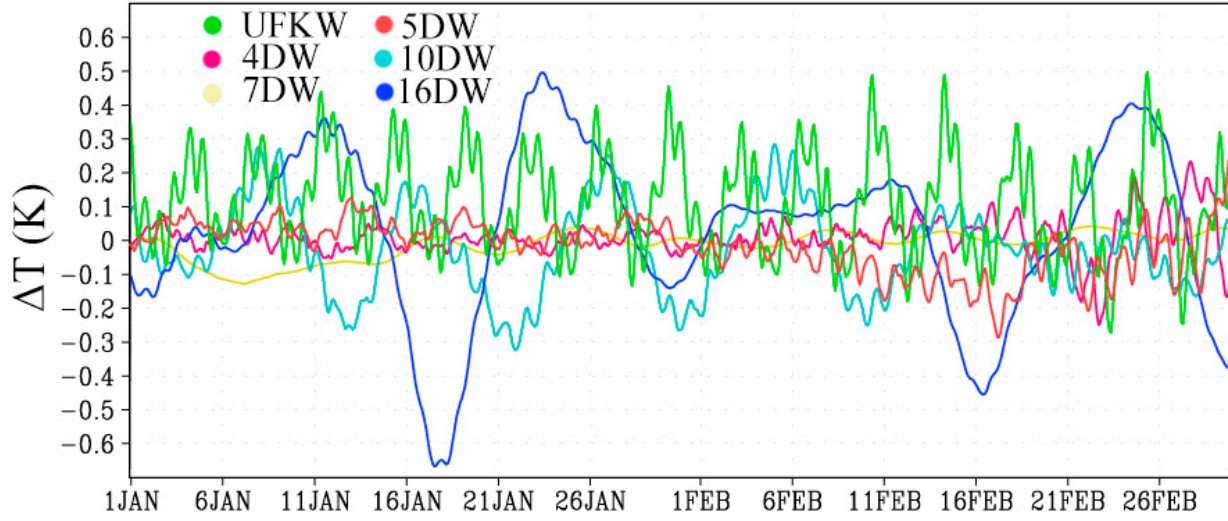

**Figure 4. Time series of mean zonal temperature variations due to the inclusion of tropospheric sources of various PW in the regions**
**of their maximum amplitudes in the MUAM.**

240  The relatively weak increments, examined in Fig. 2 and 3, require an assessment of statistical significance. Such an
241 assessment was carried out using the Student's paired t-test applied to 45312 pairs of samples in each of the latitude-altitude
grid node (64 longitude points × 708 time points for January-February with a 2-hour model output). Statistically insignificant
increments at the 95% significance level are marked with shading. In Fig. 4 shading indicates statistically insignificant data on
either temperature or RMC.

245  For a more detailed analysis of the PW effects on atmospheric circulation, the time series of zonal-mean temperature
and zonal wind variations due to the considered PW effects were observed – Fig. 4 and 5, respectively. Latitudes and heights
corresponding to the maxima of the PW amplitudes were selected: the equator, 100 km is for the UFKW; 5-day wave is
considered at 50° S and 105 km; 7-, 10- and 16-day waves: 50° N and 55 km; 4-day wave: 45° S and 105 km.

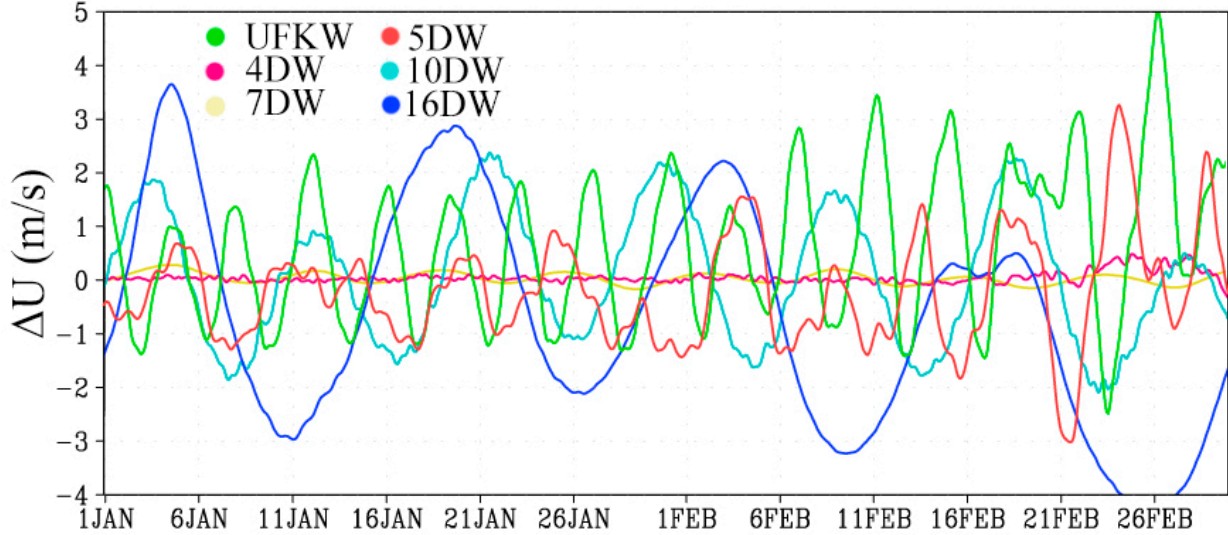

**Figure 5. Time series of zonal-mean zonal wind variations due to the inclusion of tropospheric sources of various PW in the regions of their maximum amplitudes in the MUAM.**

In all cases, especially for the zonal wind (Fig. 5), the wave structure of increments with a period corresponding to the period of the considered PW is observed. In particular, wind changes, which significantly exceed the averaged data for January and February (presented in Fig. 3) can be seen in this figure. Specifically, the inclusion of 16-day wave and the UFKW can cause the wind speed changes up to 4 m/s, and up to 5 m/s, respectively. PWs with zonal number 2 (4- and 7-day) make much smaller changes to the zonal flow, while, the weakening of the zonal flow is accompanied by the increase of these waves, as well as the 5-day wave and the UFKW by the end of February. Temperature variations in Fig. 4 have a more complex structure since temperature variations are affected not only by pressure fluctuations, but also by meridional circulation fluctuations.

**5 Conclusion and summary**

A number of model simulation have been carried out for January-February, using a 3-dimensional nonlinear mechanistic numerical model of the general circulation of the middle and upper atmosphere MUAM, to estimate the sensitivity of the atmosphere dynamic and thermal regime to the various planetary waves impact. The MUAM model allows to include selectively sources of various PW modes, which gives the opportunity to deeper study the contribution of each PW to the atmospheric circulation structure. Moreover, for a more detailed diagnostics of the PW effect on the mean flow, the transformed Eulerian mean approach was used, implying the calculation of the residual mean meridional circulation, which is a superposition of eddy and advective mean transport.

The amplitudes of the simulated PWs are consistent with the ground-based, satellite observations data, as well as with

the reanalysis and assimilation of meteorological data. The obtained increments of hydrometeorological parameters are maximal, as a rule, in the regions of maximum amplitudes of the considered PWs. In particular, the inclusion of 5-day PW and an UFKW can transform the components of the residual meridional circulation up to 6% each forming a significant contribution to the atmospheric circulation. The impact of the 16-day wave on the circulation is comparable in value with 5-day PW and UFKW, however in has different structure. Changes in circulation components occur in the high-latitude stratosphere and may reach up to 6%. In turn, all the above mentioned changes in the meridional circulation, especially its vertical component, as well as a variations of wave activity fluxes, can cause variations in the background temperature of more than 1 K. At the same time, at certain moments, this effect is much stronger. In addition, the waves can be superimposed on one another, and their effect can be summarized. I.e., the cumulative effect of the considered waves can significantly increase at certain moments of time.

The effect of 16-day PW generation by an internal atmospheric source in the southern lower thermosphere, independent of the tropospheric PW sources specified in the model, was found. Most probably, the point is that 4-day PW with a wave number 2 interacts nonlinearly with a 5-day PW with a wave number 1 causing a secondary wave excitation. Such mechanism is described, e.g., by Pogoreltsev (2001): when two waves having frequencies $\omega$ and zonal numbers $m$ interact, a new (secondary) wave arises, in which the frequency and wave number are the sum or difference of the corresponding values of the primary waves. Hence, the direct effect of the PWs can be enhanced due to their nonlinear interactions. Finally, this causes deceleration of the mean flow, creating better conditions for the SSW onset (e.g., Pogoreltsev et al., 2014). However, additional calculations are required to confirm this theory.

In addition, it should also be noted that for proper modelling of large-scale atmospheric dynamics, all models of the general atmospheric circulation should be tested for the ability to reproduce the global resonant properties of the atmosphere (the so-called atmospheric normal modes). This possibility has been repeatedly described in MUAM (e.g., Pogoreltsev, 2007, Koval et al., 2021), which underlines the reliability of the results obtained.

**Author Contributions**: All authors have made valuable contributions in writing and editing of the manuscript, data analysis and visualization of the results. A.V.K.: conceptualization, RMC calculation, writing the final version of the manuscript; T.O.N., M.A.M.: numerical modelling; T.S.E. and K.A.D.: statistical processing; G.N.M.: consulting, English editing; E.V.R.: consulting, reanalysis data processing. All authors have read and agreed to the published version of the manuscript.

**Acknowledgements**. Calculations and interpretation of the residual mean circulation, statistical analysis are supported by the Russian Science Foundation (grant # 20-77-10006). MUAM adjustment, performing numerical simulations of the atmospheric global circulation and calculation of PW structures are supported by the Ministry of Science and Higher Education of the Russian Federation (agreement 075-15-2021-583). All figures in this study are made using Grid Analysis and Display System (GrADS), which is a free software developed thanks to the NASA Advanced Information Systems Research Program.

**Availability of data and materials.** According to the statement 1296 of the Civil Code of the Russian Federation, all rights on the MUAM code belong to the Russian State Hydrometeorological University (RSHU). To get access to the codes and for their usage a reader should get a permission from the RSHU Rector at the address 79, Voronezhskaya street, St. Petersburg, Russia, 192007, phone: 007 (812) 372-50-92. The authors will assist in getting such permission. All data sets presented in the paper can be obtained from the corresponding author (Andrey Koval, a.v.koval@spbu.ru) upon request.

**Competing interests.** The authors declare that they have no competing interests.

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
