# Peer review of "Numerical modeling of relative contribution of planetary waves to the 1"

_Atmospheric Chemistry and Physics, 2022_

## Author Response (AR1)

**Author's response to Reviewer 1**

**We are grateful to the Reviewer 1 for the useful comments and suggestions. We addressed all of them. Our comments are given in bold after "//" below.**

This paper demonstrates that the contributions of individual UFKWs and PWs to changes in zonal-mean temperatures and zonal winds throughout the atmosphere are insignificant, i.e., a negative result. All other aspects of the paper in terms of approach and interpretation are very good, and I suppose it is useful to know that these waves have insignificant impacts on the mean state of the atmosphere; hence my "fair" instead" of "low" rating under "Scientific significance" above. However, I do not see the justification for a whole paper appearing in the open literature to report this result. Based on my knowledge of their other works, these authors have a great model, great ideas and great scientific insights, and I am sure that they will find a way to introduce a few sentences in one of their forthcoming papers to report this result, as an aside to the main theme of that paper.

**// First of all, on behalf of the co-authors, I would like to thank the reviewer for the positive feedback about our scientific team, as well as about the structure of the manuscript. Indeed, we still have many new ideas and we are always open for cooperation!**

**As for the reviewer's main remark, we do not quite agree with the assessment of "insignificant" and "negative".**

**We apologize to the reviewers, we have corrected Fig. 2. Due to an error in the GRADS script Fig. 2c-f presented the same arrows showing the RMC. Now everything is correct. Even if we look at the two-month average wind and RMC distributions, we can see that the statistically confirmed contribution of some waves to circulation changes can reach several percent. At the same time, at certain moments, this effect is much stronger, as shown in Figs. 4 and 5. In addition, the waves can be superimposed on one another, and their effect can be summarized. I.e., the cumulative effect of the considered waves can significantly increase at certain moments of time. This conclusion, although it looks quite obvious, is documented by us for the first time. We can articulate this thought more clearly in the final version of the manuscript.**

**We have added a file with supplementary materials to the revised version of the manuscript. There are figures depicting field changes (U, V\*, W\*) as a percentage. In particular, they show the maximum change in circulation due to the UFKW in the lower tropical thermosphere and due to the 5-day PW at the same heights in the southern hemisphere. We have included a more detailed description in the text.**

**We believe that it would be interesting to present such a study to a wide scientific community as separate paper: the most simplified statement of the problem, with clear focus only on the contribution of the PWs.**

**Author's response to Reviewer 2**

**We are grateful to the Reviewer 2 for the useful comments and suggestions. We addressed all of them. Our comments are given in bold after "//" below.**

The paper by Koval et al. aims to quantify the effect of various normal modes on the zonal wind and on the general mean circulation. For this it uses a composition of a few selected and idealized normal modes. With this the model reproduces a realistic state of the mean wind and the circulation. Essentially all Rossby modes (except 16 day) cause the same pattern. While the exercise is intellectually interesting, it remains unclear to me what its value is for the real atmosphere. The model resolution is relative coarse, the effects of GWs not well described and the waves very idealized. There should be also quite a few modes missing. Accordingly, do you want to claim that you have determinded a realistic relative measure of different modes taking into account all important effects. Then I would have my doubts. Or that you can exchange wave modes? That there is kind of a universal action for all the Rossby modes? That would be indeed something you can better show with the highly idealized setup you are using here than with something that includes all processes at once but with little control. The 16day wave is interesting and could be discussed in more detail. Overall, I think the motivation and focus of the paper needs to be sharpened before its acceptance for ACP.

**// Thanks for the helpful comments. Unfortunately, there is no universal way to study the impact of all Rossby waves, each wave has its own characteristics, depending, in particular, on the season, the impact of large-scale processes such as QBO, ENSO, etc. Therefore, we have chosen only a part of the spectrum of planetary waves, the amplitudes of which are maximized during the boreal winter. Below we have responded to all comments in order. We have also extended the description of the 16-day wave effect.**

Specific comments:

L62 Please substantiate this claim: What makes the model "modern and promising"? The spatial resolution is coarse. Please compare to early work by Kevin Hamilton (resolution needed for tropics/extratropics) to check whether it is sufficient.

**// The spatial resolution of our model is quite coarse, however, as our previous studies have shown, this resolution is more than enough to resolve global atmospheric oscillations, including tides (e.g., Suvorova & Pogoreltsev, 2011; Shevchuk et al. 2018; Didenko et al., 2021) and planetary waves (e.g., Gavrilov et al., 2018; Koval et al., 2018a; 2018b; 2019; 2022). The discussion was added to the text.**

L74 GWs are one of the most important drivers of the atmosphere. Hence there needs to be a description/reference to the actual state of the pramaterization. For this the paper refers to Koval et al., JGR 2022, but there is nothing either! Please include some description of the parametrizations in this paper. Could be also in an appendix.

**// Indeed, GWs are very important for the correct reproduction of the atmospheric circulation. We have not mentioned the GW accounting in the MUAM in this article in detail, because the paper is dedicated to the study of planetary waves. Naturally, the GW (of orographic and non-orographic origin) cannot be resolved by the model, so parameterizations are used to take them into account. There are three of them in MUAM. We have added an appropriate description to the paper.**

L88 In this way you have very idealized waves matching perfectly the Eigenmode shape. Which consequences does that have for the breaking of the waves and the altitude and location where they depsosit their momentum. Also, limiting yourself to zonal wavenumbers 1 and 2 is very idealized.

**// We do not quite understand what is meant by "idealized" In this context. The sources of westward propagating PWs (the so-called free oscillations, or normal modes) are specified in MUAM so that they correspond to oscillations observed in the atmosphere (e.g., Pogoreltsev et al., 2009; Koval et al., 2018b).**
**In all our research papers, usually we pay special attention to the study of the wave - mean flow interactions. For this purpose, we typically analyze the refractive index of the atmosphere, the Eliassen-Palm flux and its divergence, and also analyze changes in the residual meridional circulation, which is partly driven by wave action (the so-called Downward Control Principle). In the MUAM there are sources of PWs with zonal wavenumbers 3 (e.g. quasi-two-day wave). However, as you can see from the current paper, even PWs with wavenumber 2 play a very weak role compared to the zonal number 1. With an increase in the zonal wavenumber, the effect weakens, especially since the maximum amplitude of the 2-day wave is usually observed in the boreal summer months, so they remained outside the scope of our study.**

**We have extended description of wave sources and added discussion of wave-mean flow interactions to the article.**

Technical corrections:

L27 gramitically which refers to circle: please restructure senetence

**// corrected**

L122 uneven -> different for different panels

**// corrected**

L147 MLS ? Please be precise

**// corrected**

**References:**

Didenko K.A., Pogoreltsev A.I., Koval A.V., Ermakova T.S. (2021) Investigation of solar thermal tides using model data // Proc. SPIE 11916, 27th International Symposium on Atmospheric and Ocean Optics: Atmospheric Physics, 1191687. doi: 10.1117/12.2603432
Gavrilov, N.M., Koval, A.V., Pogoreltsev, A.I., Savenkova, E.N. 2018. Simulating planetary wave propagation to the upper atmosphere during stratospheric warming events at different mountain wave scenarios // Advances in Space Research. V. 61, I. 7, p. 1819–1836 doi 10.1016/j.asr.2017.08.022
Koval, A.V., Gavrilov, N.M., Pogoreltsev, A.I., Savenkova, E.N. 2018a. Comparisons of planetary wave propagation to the upper atmosphere during stratospheric warming events at different QBO phases // Journal of Atmospheric and Solar-Terrestrial Physics. V. 171, P. 201-209. Doi 10.1016/j.jastp.2017.04.013
Koval, A. V., Gavrilov, N. M., Pogoreltsev, A. I., & Shevchuk, N. O. 2018b. Influence of solar activity on penetration of traveling planetary-scale waves from the troposphere into the thermosphere. Journal of Geophysical Research: Space Physics, 123. (8), 6888-6903, doi: 10.1029/2018JA025680
Koval, A. V., Gavrilov, N. M., Pogoreltsev, A. I., & Shevchuk, N. O. 2019. Reactions of the middle atmosphere circulation and stationary planetary waves on the solar activity effects in the

thermosphere. Journal of Geophysical Research: Space Physics, 124, 10645-10658 doi: 10.1029/2019JA027392

Koval, A.V., Gavrilov, N.M., Kandieva, K.K. Ermakova, T.S., Didenko, K.A. (2022) Numerical simulation of stratospheric QBO impact on the planetary waves up to the thermosphere // Scientific Reports, 12, 21701. DOI: 10.1038/s41598-022-26311-x

Pogoreltsev, A. I., Kanukhina, A. Yu., Suvorova, E. V., Savenkova, E. N. (2009). Variability of Planetary Waves as a Signature of Possible Climatic Changes. J. Atmos. Solar-Terr. Phys. 71: 1529-1539. doi:10.1016/j.jastp.2009.05.011

Shevchuk N. O., Ortikov M. Yu., Pogoreltsev, A. I. Modeling of atmospheric tides with account of diurnal variations of ionospheric conductivity. Russian Journal of Physical Chemistry B. 2018, vol. 12, no. 3, pp. 576–589. DOI: 10.1134/S199079311803017X.

Suvorova, E. V., Pogoreltsev, A. I., 2011. Modeling of nonmigrating tides in the middle atmosphere // Geomagmetizm and Aeronomy. 51(1), 105-115. DOI: 10.1134/S0016793210061039

**Andrey Koval, on the behalf of co-authors.**